# Analysis of Net Primary Productivity of Retired Farmlands in the Grain-for-Green Project in China from 2011 to 2020

Yuanming Xie [1,†], Zemeng Ma [1,†], Mingjie Fang [1], Weiguo Liu [2,3], Feiyan Yu [2], Jiajing Tian [2], Shuoxin Zhang [1,3,*] and Yan Yan [1,3,*]

1   College of Forestry, Northwest Agriculture and Forestry University, Yangling 712100, China
2   Center for Ecological Forecasting and Global Change, College of Forestry, Northwest Agriculture and Forestry University, Yangling 712100, China
3   Qinling National Forest Ecosystem Research Station, Huoditang, Ningshan 711600, China
*   Correspondence: sxzhang@nwsuaf.edu.cn (S.Z.); yanyanemail@nwafu.edu.cn (Y.Y.)
†   These authors contributed equally to this work.

**Abstract:** The Grain-for-Green Project (GFGP), one of the largest ecological restoration projects in China, has made a significant contribution to carbon neutrality. However, the quantitative contribution to climate change and the driving forces of the carbon sequestration of retired farmlands remains unclear. To analyze the carbon dynamics of the retired farmlands and their driving forces, GlobeLand30 databases were used to identify retired farmlands from 2001 to 2020; in addition, net primary productivity (NPP) of the identified lands was estimated with the Carnegie–Ames–Stanford Approach (CASA). Results showed that 131,298 km$^2$ of farmlands were retired from 2001 to 2010 (L01–10), and 130,225 km$^2$ were retired from 2011 to 2020 (L11–20). The largest areas of retired farmlands were South Central China (24%) in L01–10 and Northwest China (22%) in L11–20. The annual NPP increment of retired farmlands was the highest and most significant in Southwest China (26,455–28,783 GgC·year$^{-1}$ for retired farmlands in L01–10 and 21,320–23,303 GgC·year$^{-1}$ in L11–20). In this study, NPP had significantly positive correlations with temperature and precipitation as well as significant constraint relationships with rural population density and animal husbandry output value. The findings could provide suggestions for the further implementation of the GFGP and other restoration projects.

**Keywords:** carbon sequestration; driving force; CASA model; land use change; ecological restoration project

## 1. Introduction

In the context of global climate change and efforts to decarbonize, enhancing carbon sinks in terrestrial ecosystems can provide an important approach to mitigate increases in anthropogenic atmospheric $CO_2$ and achieve carbon neutrality targets. Many ecological restoration projects have been launched around the world that can serve this purpose, such as the Conservation Reserve Program and the Grain-for-Green Project (GFGP). Germany and the International Union for Conservation of Nature initiated the Bonn Challenge in 2011 to restore degraded and deforested lands [1]. The Conservation Reserve Program (CRP) enacted by the U.S. government in 1986 has slowed soil erosion and improved wildlife habitats [2]. The Chinese government enacted the policy of the Grain-for-Green Project in the late 1990s to reduce soil erosion and desertification by returning farmlands prone to soil erosion with low productivity to forest and grassland habitats [3]. All of these ecological restoration projects have exerted significant carbon storage benefits by revegetating degraded lands [4,5]. Afforestation and ecological restoration are considered to be important drivers of terrestrial carbon sinks in China [6]. The implementation of the GFGP resulted in over 335,000 km$^2$ of new afforestation with high carbon sequestration potential, with a cumulative carbon sink reaching 1697 TgC in 2020 [7,8].

The net primary productivity (NPP) of vegetation is a commonly used indicator to evaluate the carbon storage benefits of ecological restoration projects, defined as the difference in carbon stock between photosynthesis and autotrophic respiration [9]. To estimate NPP at large scales, field and laboratory measurements are not feasible because of high costs and a lack of representative field data [10]. With advances in remote-sensing technology, the estimation of NPP at regional to global scales has been greatly improved by the large size of samples from satellites and the use of model simulation [11–13]. The estimation of NPP using remote-sensing data facilitates the analysis of the spatial and temporal variation in the NPP of these ecological restoration projects and associated factors influencing NPP. Among them, light use efficiency models (e.g., the Global Production Efficiency Model (GLO-PEM) [14] and CASA models [15]) can be used to obtain the spatial and temporal characteristics of NPP with the use of remote-sensing data, and thus compensate for the labor-intensive plot-based measurements of regional studies and their related limitations.

With an accurate estimation of NPP, the identification of driving forces that affect it could assist in developing better management practices for restoration projects to enhance carbon storage benefits and other ecological services [16]. Based on previously published studies, climate change, human activities, ecosystem types, and restoration policies influence the NPP in restoration areas [17,18]. Climate change has a direct impact on vegetation growth through changes in temperature, precipitation, and light [19,20]; human activities act directly or indirectly on vegetation growth through different types of disturbances [21,22], and there are complex interactions and response mechanisms between the two at spatial and temporal scales. Wu et al. [23] quantitatively estimated the impact of anthropogenic factors on NPP using population and GDP data. Wang et al. [24] analyzed the effect of the implementation of the GFGP on the trade-offs among ecosystem services such as NPP, soil conservation, and water yield in the upper reaches of the Yangtze River. Nonetheless, most previous studies have focused on local and regional (watershed) scales [25–27], with anthropogenic influences used to focus on land-use type conversion and regional economic development [22,28]. Fewer studies have been conducted to quantify the effects of specific human activities on vegetation NPP for large-scale ecological restoration projects.

Many studies have been conducted in China on the GFGP, which, as one of the most ambitious restoration projects, has provided significant carbon storage benefits [29]. For example, Deng and Shangguan [30] investigated the dynamics of carbon sequestration during forest restoration in Shaanxi Province based on forest inventory data and empirical factors. Feng et al. [29] used remote sensing and ecological modeling to analyze the changes in carbon sequestration that occurred after the implementation of the GFGP on the Loess Plateau. However, the GFGP was implemented nationwide with high spatial heterogeneity in both natural and social factors [31]. Small-scale studies have difficulty distinguishing the main factors involved in carbon storage in different areas. The identification of the driving forces for the NPP in the areas in which the GFGP has been implemented can provide suggestions for further development of the GFGP and other possible restoration projects. Therefore, the main objectives of this study were to: (1) reveal the spatial pattern of retired farmlands resulting from the implementation of the GFGP in China from 2000 to 2020; (2) assess the spatial and temporal variations in the NPP in the retired farmlands from 2011 to 2020; and (3) identify the driving forces affecting NPP in the retired farmlands.

## 2. Materials and Methods

### 2.1. Identification of Retired Farmlands from GFGP

The Chinese government expanded the GFGP from Sichuan, Shaanxi, and Gansu provinces to include 174 counties in 13 western administrative regions in 2000, and then the project was further expanded to the entire country in 2002 [7]. Large areas of farmland have been retired in the last two decades. The retired farmlands were identified by the global land use database GlobeLand30 v2000, v2010, and v2020 (http://www.globallandcover.com/

accessed on 13 December 2020), which was provided by the National Geomatics Center of China and has high classification accuracy. The classification accuracy of farmlands, forests, shrublands, and grasslands was 83, 84, 72 and 73%, respectively [32]. In 2000, farmland was considered retired if it had been converted to other terrestrial vegetation in 2010 and 2020. The retired farmlands were resampled to 900 m resolution after being identified from the GlobeLand30 database.

To better reveal the changes in NPP and the driving factors in the retired farmlands, China was previously divided into six major regions (Resource and Environment Science and Data Center, https://www.resdc.cn/ accessed on 25 November 2021): Northeast China (NE), South Central China (SC), East China (E), North China (N), Northwest China (NW), and Southwest China (SW) based on differences in climate conditions, soils, and topographic characteristics (Figure 1); the present study uses these regions. The retired farmlands were divided into two groups according to the year of land conversion as follows: lands retired between 2001 and 2010 (L01–10) and lands retired from 2011 to 2020 (L11–20).

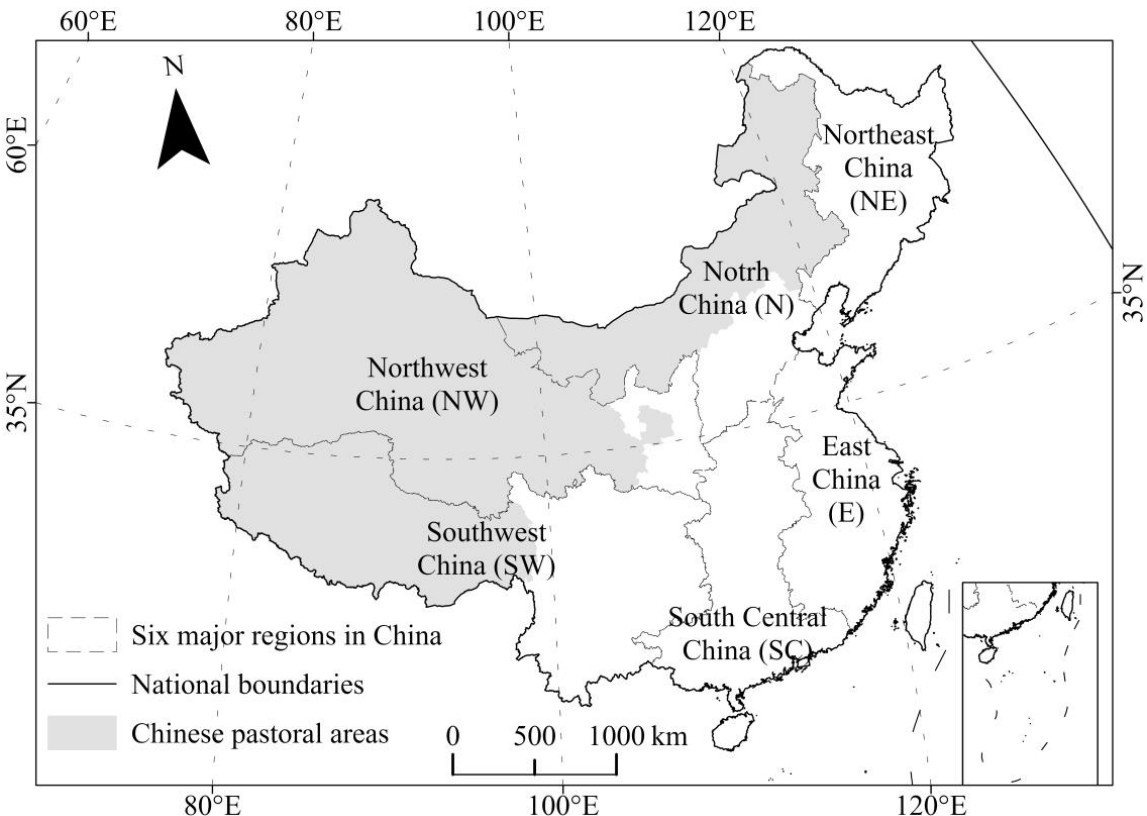

**Figure 1.** Six major regions and the major pastoral areas in China.

### 2.2. NPP Estimation for the Retired Farmlands

The NPP of the retired farmlands for each grid cell was estimated by the Carnegie–Ames–Stanford approach (CASA) model. The CASA model was developed by Potter et al. [15] and has been intensively validated and widely used in the Americas, as well as in Eurasia [33,34]. The inputs of this model are typically vegetation type, monthly normalized difference vegetation index (NDVI), monthly solar radiation, mean monthly air temperature, and monthly precipitation. The *NPP* is determined by two variables, photosynthetically active radiation absorbed by the vegetation (*APAR*) and actual light use efficiency ($\varepsilon$) [32,35]:

$$NPP(x, t) = APAR(x, t) \times \varepsilon(x,t) \tag{1}$$

where for grid cell $x$ in month $t$, $NPP(x, t)$ represents the net primary productivity (gC·m$^{-2}$), $APAR(x, t)$ is the photosynthetically active radiation absorbed by the vegetation (MJ·m$^{-2}$), and $\varepsilon(x, t)$ is the actual light use efficiency (gC·MJ$^{-1}$).

The photosynthetically active radiation was calculated based on solar radiation and NDVI. The monthly solar radiation was obtained from the China Meteorological Data Service Center (http://data.cma.cn/ accessed on 29 October 2021) and interpolated to 900 m resolution in ArcGIS ver. 10.8 using ordinary Kriging and inverse distance weighting. The NDVI data (250 m × 250 m resolution) from 2011 to 2020 (MOD13Q1) were obtained from the moderate-resolution imaging spectroradiometer (MODIS) and products provided by the United States National Aeronautics and Space Administration (https://www.nasa.gov/ accessed on 12 August 2022). The MOD13Q1 data were pre-processed and stitched to the national scale using the MODIS Reprojection Tool and were used to generate monthly NDVI by the Maximum Value Composite procedure. The NDVI data were then resampled to 900 m resolution.

The actual light use efficiency of each vegetation type was calculated based on the maximum light use efficiency of each vegetation type and the local climate. The maximum light use efficiency for each vegetation type used here was the well-validated parameter by Zhu et al. [36]. The efficiencies for the evergreen coniferous forest, evergreen broadleaf forest, deciduous coniferous forest, deciduous broadleaf forest, mixed forest, shrubland, grassland, and cultivated land were 0.389, 0.985, 0.485, 0.692, 0.475, 0.429, 0.542, and 0.542 gC·MJ$^{-1}$, respectively. The vegetation type (MCD12Q1) was also obtained from the MODIS. The mean monthly air temperature and monthly precipitation were obtained from the China Meteorological Data Service Center and interpolated to 900 m resolution.

## 2.3. Analysis of Spatial and Temporal Variation in NPP

The trends of variation in NPP from 2011 to 2020 for each grid cell were analyzed by the Sen trend degree method. This method does not require a normal distribution of time series data and is resistant to noise interference in comparison to linear regression [37,38]. The significance tests for the trends of the variation in NPP were conducted by the Mann–Kendall trend test method. Equation (2) was used to calculate the Sen trend degree ($\beta$). A positive or negative value of $\beta$ indicates that the NPP had an increasing or decreasing trend over time, respectively:

$$\beta = \text{Median}\left(\frac{x_j - x_i}{j - i}\right), \forall j > i \tag{2}$$

where $x_i$ and $x_j$ are the NPP of two consecutive years $i$ and $j$.

The values of the Sen trend degree ($\beta$) and the Mann–Kendall trend test ($Z$) for each grid cell were obtained by the raster calculation package *raster* and the non-parametric trend test package *trend* in R software. According to the values of $\beta$ and $Z$, the variation trends in vegetation NPP were classified into nine levels: −4, −3, −2, −1, 0, 1, 2, 3, and 4 (Table S1).

## 2.4. Analysis of Driving Forces for NPP Variation

Many factors affect the NPP of vegetation. In this study, the effect of climatic factors (i.e., annual mean air temperature and precipitation) and anthropogenic factors (i.e., rural population and animal husbandry) on NPP were analyzed. The indicators for the rural population and animal husbandry were the rural population density and the output value of animal husbandry, respectively.

The annual mean air temperature and precipitation from 2011 to 2020 were calculated for each prefectural-level city based on the mean monthly air temperature and monthly precipitation data from the China Meteorological Data Service Center. The rural population density and the output value of animal husbandry from 2011 to 2020 for each prefectural-level city were derived from provincial and municipal yearbooks (National Digital Electronic Library, http://www.nlc.cn/ accessed on 2 June 2022). In this paper, the concept of a province often includes other large administrative areas such as autonomous

regions including Tibet and Xinjiang. The output values of animal husbandry were retrieved only for the Chinese pastoral areas that include Xinjiang, Tibet, Qinghai, Gansu, and Inner Mongolia (Figure 1) due to the high frequency of open grazing in unfenced areas that occurs in those areas [39].

To quantify the effect of the factors on the vegetation NPP, the indicators for the factors were analyzed separately for six different regions (NE, SC, E, N, NW, and SW). The Pearson correlation coefficients (*r*) between the factors and NPP were calculated for each region. Traditional correlation analysis and regression methods focus on the relationship between the mean values of variables, which is less likely to reflect the complexity of ecological processes. Meanwhile, because the relationship between human activities and vegetation NPP is not simply one-to-one, it is difficult to explain the relationship between them with traditional methods. The constraint line method focuses on the data boundary between two variables in a complex ecological process, and it is easier to use to find the causal relationship and limiting effect between variables than the traditional methods, so the constraint line method was chosen to analyze the relationship between human activities and NPP [40]. To calculate the constraint relationships for each region, the datasets were divided into 100 subgroups equally, and the 99% quantile of each subgroup was selected as the boundary point for least-squares fitting. The statistical analysis and necessary program were written in R software (version 4.2.0).

## 3. Results

### 3.1. Spatial Distribution Patterns of Retired Farmlands

During the implementation of the GFGP in 1999, the major target regions were the northeastern mountains (NE), the Yunnan-Guizhou Plateau (SW), and the hilly areas in the middle and lower reaches of the Yangtze River (E). From 2001 to 2010, the total area of retired farmlands was 131,298 km$^2$, of which 70,694 km$^2$ was converted to forests and shrublands, and 60,604 km$^2$ was converted to grasslands (Table 1). Most of the retired farmlands were in SC (24%), followed by SW (22%) and N (20%). Those three regions accounted for more than 66% of the total retired farmlands in this decade (Figure 2). When the retired farmlands were upscaled to the county level, the counties with more square kilometers of farmlands converted to grasslands were mainly concentrated in the NE, N, and the eastern part of the NW region, whereas the counties with more square kilometers of farmlands converted to forests and shrublands were mainly concentrated in the SW, SC, and western parts of the NE (Figure S1a,c).

**Table 1.** Area of retired farmlands in the six regions from 2001 to 2020: km$^2$.

| Group of Retired Farmlands | Land Conversion Type | NE | N | E | NW | SW | SC |
|---|---|---|---|---|---|---|---|
| L01–10 | To forests and shrublands | 6248.34 | 3969 | 10,818.36 | 4744.17 | 18,509.31 | 26,404.38 |
| | To grasslands | 8712.36 | 22,658.94 | 3408.48 | 11,101.86 | 9725.67 | 4996.89 |
| L11–20 | To forests and shrublands | 11,634.03 | 3232.71 | 15,419.97 | 4361.85 | 20,088.81 | 21,274.65 |
| | To grasslands | 9579.06 | 17,849.97 | 3986.82 | 9901.44 | 8803.89 | 4092.12 |

Note: NE, Northeast China; NW, Northwest China; SW, Southwest China; SC, South Central China; E, East China; N, North China; L01–10, lands retired between 2001 and 2010; L11–20, lands retired between 2011 and 2020.

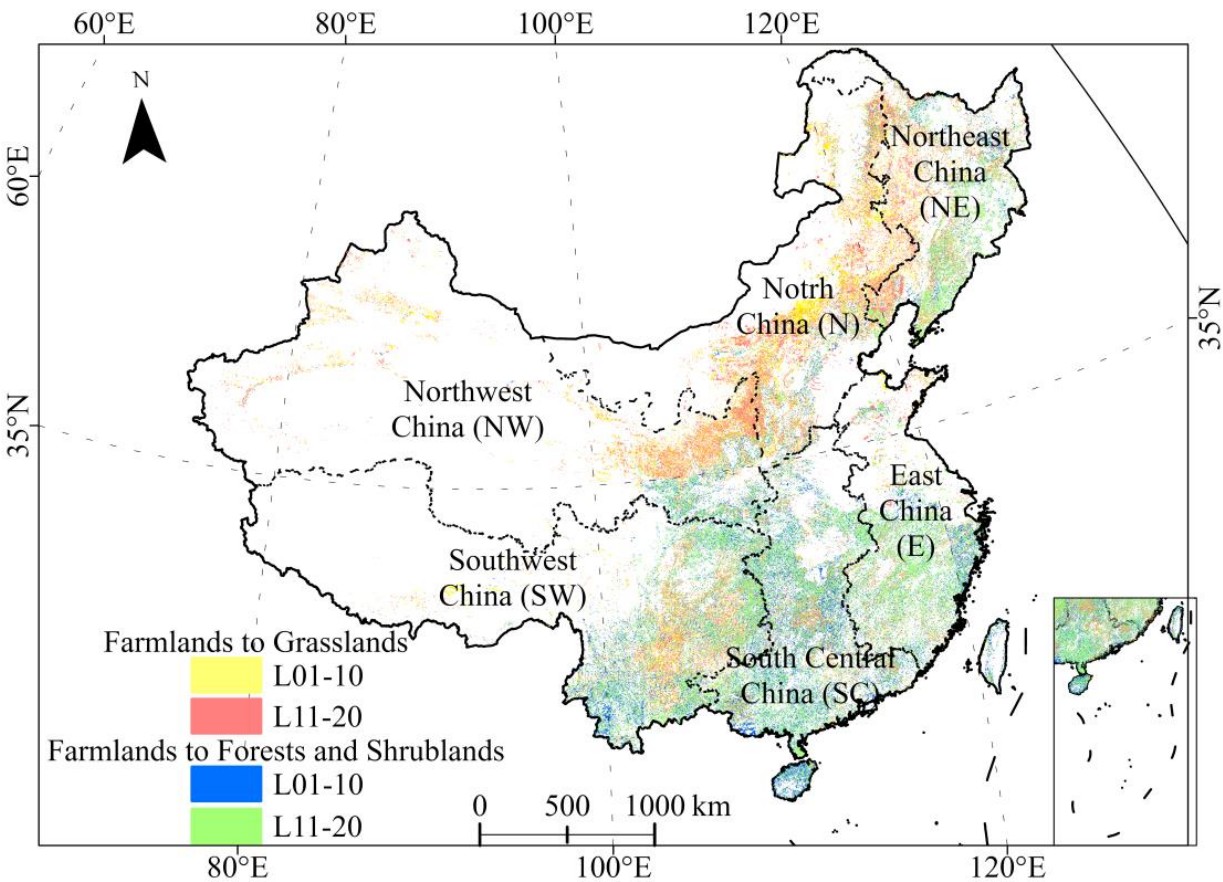

**Figure 2.** Spatial distribution of retired farmlands converted to grasslands as well as to forests and shrublands in the Grain-for-Green Program. Note: L01–10, lands retired between 2001 and 2010; L11–20, lands retired between 2011 and 2020.

In the last decade, from 2011 to 2020, the total spatial extent of retired farmlands was 130,225 km$^2$. Among the retired farmlands, 58% were converted to forests and shrublands (Table 1). Most of the retired farmlands were in the NW region (22%), followed by the SC (20%), NE (16%), and N (16%) regions. The NW and SC accounted for more than 42% of the total retired farmlands in this decade. At the county level, the overall pattern of land conversion was roughly consistent from 2001 to 2010, but the number of counties that converted farmlands to forests and shrublands increased (Figure S1b,d).

*3.2. Spatial and Temporal Variation in NPP on Retired Farmlands*

By comparing the area-specific NPP in the six regions (Table 2), SW had the highest area-specific NPP for L01–10 and L11–20, followed by SC and E. The lowest average NPP was found in N for L01–10 and L11–20. Between the two groups of retired farmlands (L01–10 and L11–20), the average NPP of L11–20 in the E and NE regions was slightly higher than that of L01–10, and the NPP of L11–20 in the NW, SW, and N regions was slightly lower than that of L01–10 (Table 2).

**Table 2.** Average values of net primary productivity (NPP) for vegetation on fallow land in six regions from 2011 to 2020.

| Group of Retired Farmlands | Year | Annual NPP (gC·m$^{-2}$·Year$^{-1}$) | | | | | |
|---|---|---|---|---|---|---|---|
| | | NE | NW | SW | SC | E | N |
| L01–10 | 2011 | 632.56 | 534.73 | 890.34 | 842.80 | 822.43 | 491.12 |
| | 2012 | 691.51 | 578.58 | 841.94 | 841.66 | 855.53 | 551.59 |
| | 2013 | 673.97 | 584.40 | 917.12 | 872.12 | 826.74 | 580.40 |
| | 2014 | 683.32 | 588.37 | 895.52 | 855.03 | 844.99 | 561.12 |
| | 2015 | 692.02 | 579.25 | 972.97 | 913.92 | 889.83 | 541.28 |
| | 2016 | 689.01 | 580.68 | 963.43 | 913.25 | 866.39 | 573.25 |
| | 2017 | 693.13 | 585.00 | 941.29 | 899.10 | 867.77 | 563.44 |
| | 2018 | 681.28 | 616.57 | 938.70 | 916.97 | 920.11 | 568.71 |
| | 2019 | 730.02 | 623.79 | 939.64 | 855.52 | 829.80 | 605.34 |
| | 2020 | 787.22 | 648.10 | 883.01 | 875.23 | 846.06 | 547.49 |
| L11–20 | 2011 | 656.70 | 523.30 | 880.76 | 845.32 | 840.12 | 477.77 |
| | 2012 | 720.03 | 573.02 | 833.98 | 840.83 | 876.69 | 546.32 |
| | 2013 | 700.70 | 578.10 | 921.68 | 872.28 | 837.28 | 562.85 |
| | 2014 | 707.25 | 583.15 | 893.28 | 859.53 | 877.12 | 549.81 |
| | 2015 | 724.36 | 570.37 | 976.22 | 919.06 | 912.73 | 530.13 |
| | 2016 | 726.10 | 571.05 | 963.10 | 917.86 | 886.71 | 559.53 |
| | 2017 | 712.11 | 577.13 | 941.48 | 904.67 | 893.72 | 552.44 |
| | 2018 | 706.46 | 610.40 | 947.68 | 916.33 | 942.04 | 557.08 |
| | 2019 | 763.05 | 617.93 | 947.24 | 852.65 | 845.69 | 595.65 |
| | 2020 | 802.53 | 637.82 | 891.64 | 882.56 | 873.12 | 541.29 |

Note: NE, Northeast China; NW, Northwest China; SW, Southwest China; SC, South Central China; E, East China; N, North China; L01–10, lands retired between 2001 and 2010; L11–20, lands retired between 2011 and 2020.

Among the six regions, the annual NPP increment in L01–10 was the highest in SC (26,455–28,783 GgC·year$^{-1}$), followed by the SW (23,771–27,471 GgC·year$^{-1}$) and NW (8473–10,269 GgC·year$^{-1}$), the latter of which was the lowest. In contrast, the annual NPP increment in L11–20 was the highest in the SW (24,095–28,205 GgC·year$^{-1}$), followed by SC (21,320–23,303 GgC·year$^{-1}$), and the lowest was also in NW (7464–9097 GgC·year$^{-1}$) (Table S2). The total carbon stored by the retired farmlands was 2014 TgC from 2011 to 2020.

Based on the Mann–Kendall trend test and Theil-Sen slope results, at the grid cell level, over 71 and 73% of grid cells presented increasing trends in NPP in the L01–10 and L11–20 time periods (Table 3). The percentages of grid cells with extremely significant increasing trends in NPP were 2.76 and 3.03% in the L01–10 and L11–20 grounds, respectively (Table 3). The percentages of grid cells with extremely significant increasing trends in NPP were concentrated in the northeastern mountains in NE, N, and in the Yunnan-Guizhou Plateau area in SW (Figure 3). The percentages of grid cells with an extremely significant reduction in NPP were 0.35% in L01–10 and 0.23% in L11–20 (Table 3). Those grid cells were distributed in N and in the hilly areas in the middle and lower reaches of the Yangtze River (Figure 3).

### 3.3. Factors Affecting NPP

In this study, the annual mean air temperature and precipitation were analyzed as climatic factors. In both periods for retired farmlands, NPP had positive correlations with annual mean air temperature in all six regions (Figure 4). The regression coefficients for the linear correlations can be found in the Supplementary Information (Table S3). All the climate relationships with NPP were significant except in the NE and NW regions. The coefficients of determination were higher in L11–20 than in L01–10. The strongest linear correlations were found between annual mean air temperature and NPP in SC ($R^2$ = 0.573, $p < 0.001$ for L01–10 and $R^2$ = 0.707, $p < 0.001$ for L11–20).

**Table 3.** Statistics of the change in net primary production in six major regions from 2011 to 2020.

| Group of Retired Farmlands | Trend Categories | NE | NW | SW | SC | E | N | Total |
|---|---|---|---|---|---|---|---|---|
| L01–10 | 4 | 1498 | 757 | 697 | 558 | 258 | 711 | 4479 |
| | 3 | 3474 | 2782 | 2963 | 2702 | 1097 | 2977 | 15,995 |
| | 2 | 1135 | 1068 | 1206 | 1166 | 467 | 1390 | 6432 |
| | 1 | 10,123 | 11,006 | 18,342 | 19,799 | 9044 | 19,391 | 87,705 |
| | −1 | 2151 | 3625 | 10,402 | 13,377 | 5914 | 7973 | 43,442 |
| | −2 | 29 | 88 | 357 | 357 | 196 | 168 | 1195 |
| | −3 | 49 | 185 | 713 | 645 | 402 | 228 | 2222 |
| | −4 | 8 | 51 | 177 | 148 | 147 | 31 | 562 |
| L11–20 | 4 | 1997 | 655 | 805 | 429 | 339 | 644 | 4869 |
| | 3 | 4839 | 2471 | 3285 | 2092 | 1537 | 2558 | 16,782 |
| | 2 | 1695 | 1036 | 1411 | 916 | 703 | 1143 | 6904 |
| | 1 | 14,444 | 10,069 | 19,912 | 15,967 | 12,470 | 15,632 | 88,494 |
| | −1 | 3083 | 3068 | 9433 | 11,061 | 8106 | 5688 | 40,439 |
| | −2 | 35 | 76 | 251 | 284 | 241 | 110 | 997 |
| | −3 | 80 | 172 | 475 | 467 | 452 | 219 | 1865 |
| | −4 | 7 | 62 | 97 | 87 | 81 | 34 | 368 |

Note: NE, Northeast China; NW, Northwest China; SW, Southwest China; SC, South Central China; E, East China; N, North China; L01–10, lands retired between 2001 and 2010; L11–20, lands retired between 2011 and 2020; trend categories are defined in Table S1.

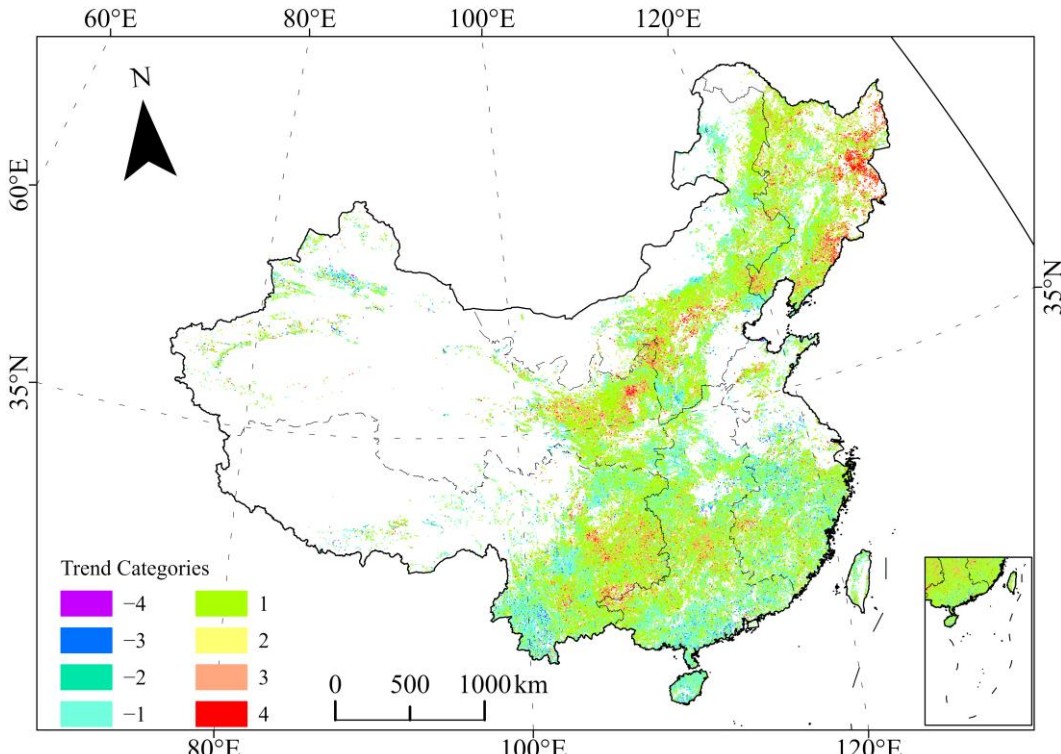

**Figure 3.** Trends in net primary production of the Grain-for-Green Program from 2011–2020. Positive value represents an increase; negative value represents a decrease; and a larger value represents an increase in significance. Trend categories are defined in Table S1.

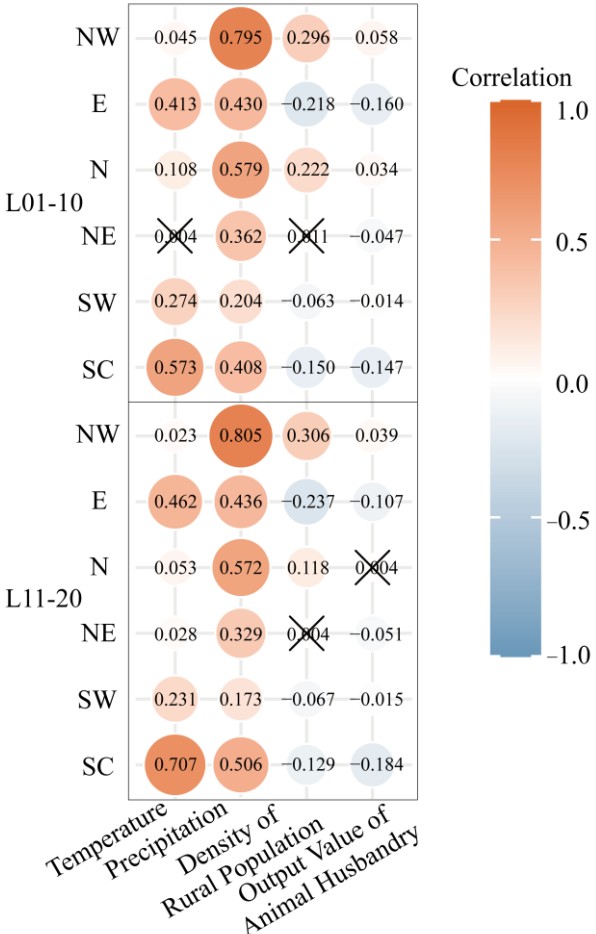

**Figure 4.** Correlation between net primary productivity (NPP) and climatic factors and anthropogenic factors. Note: NE, Northeast China; NW, Northwest China; SW, Southwest China; SC, South Central China; E, East China; N, North China; L01–10, lands retired between 2001 and 2010; L11–20, lands retired between 2011 and 2020.

For the annual precipitation, the correlations to NPP were also positive in all six regions (Figure 4). All the relationships were significant in both periods (Table S3), with the highest coefficients of determination in NW, followed by the N, SC, and E regions. The strongest linear regression models were in NW ($R^2$ = 0.795, $p < 0.001$ for L01–10 and 0.805, $p < 0.001$ for L11–20), and the weakest linear regression models were found in SW ($R^2$ = 0.204, $p < 0.001$ for L01–10 and 0.173, $p < 0.001$ for L11–20).

The rural population density and the output value of animal husbandry were analyzed as anthropogenic factors. The NPP of the GFGP and rural population density were positively correlated in the NW, N, and NE regions, and the correlations were negative in SW, E, and SC (Table S3). All the correlations were insignificant (Figure 4). The highest coefficients of determination were found in the NW region ($R^2$ = 0.296, $p < 0.001$ in L01–10, and $R^2$ = 0.306, $p < 0.001$ in L11–20). For the NPP and animal husbandry output value, the NW and N regions showed weak positive correlations, and the remaining regions showed weak negative correlations (Table S3). All the correlations were insignificant. The highest coefficients of determination were 0.16 ($p < 0.001$) in the E region in L01–10 and 0.184 ($p < 0.001$) in the SC region in L11–20 (Figure 4).

Although the linear regression could not detect a significant relationship between the anthropogenic factors and NPP, the constraint relationships were analyzed based on the highly clustered scatter points. At the national level, the influence of rural population density on NPP was confined as a quadratic line (Figure 5a,b). The constraint relationships were significant in both groups ($R^2$ = 0.655 in L01–10 and $R^2$ = 0.543 in L11–20). Strong

constraint relationships were detected between the output value of animal husbandry and the NPP of the pastoral areas in the GFGP (Figure 5c,d), and the output value of animal husbandry had a stronger constraint effect on the NPP of L11–20 ($R^2$ = 0.659) than that of L01–10 ($R^2$ = 0.605).

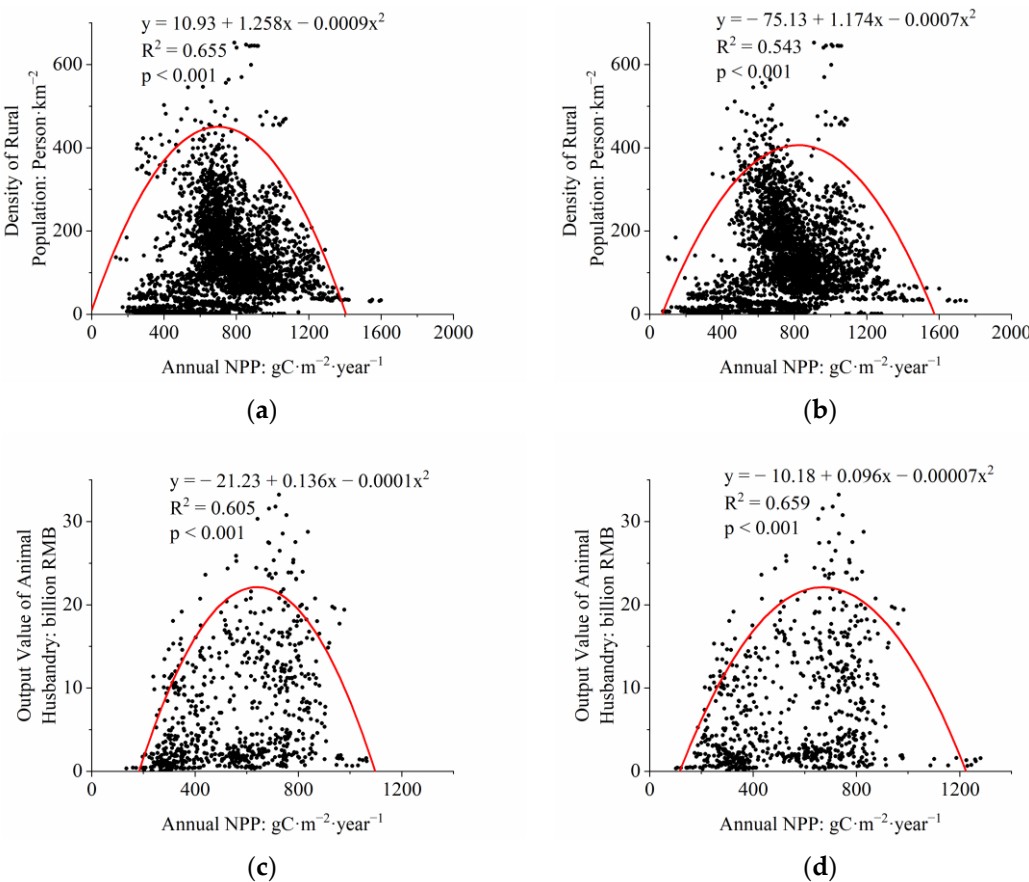

**Figure 5.** National-level constraint relationships between net primary production (NPP) and (**a**) rural population density in L01–10 and (**b**) in L11–20, as well as between NPP and animal husbandry output value in (**c**) L01–10 and (**d**) L11–20 in pastoral areas. Note: RMB, renminbi.

When the six regions were analyzed separately, the constrained relationship between rural population density and NPP was significantly higher for L01–10 than for L11–20 in all six regions except for SW (Figure S3a,b). The strongest constraint relationships were in E ($R^2$ = 0.63 for L01–10 and $R^2$ = 0.46 for L11–20), followed by SC ($R^2$ = 0.566 for L01–10 and $R^2$ = 0.432 for L11–20). The weakest constraint relationships were in SW for L01–10 ($R^2$ = 0.294) and in the NE region for L11–20 ($R^2$ = 0.211). The constraint relationships between the animal husbandry output value and NPP in pastoral areas had the highest coefficient of determination only in N ($R^2$ = 0.633 for L01–10 and $R^2$ = 0.579 for L11–20) (Figure S3c,d). The shapes of the constraint lines changed among regions.

## 4. Discussion

### 4.1. Retired Farmlands

The GFGP focuses on converting farmlands on steep slopes to forests, shrublands, and grasslands; it is also known as the Sloped Land Conversion Program [41]. Thus, large areas of retired farmlands were concentrated in the SC, SW, and NW as well as the middle and lower reaches of the Yangtze River with hilly areas with more sloping farmlands. Farmers' willingness to retire farming has changed over time. Many local farmers were very willing to retire their farmlands in the first round of the GFGP (1999–2013), because they were provided with subsidies that had the potential to improve their livelihoods; a

considerable portion of the farmers planned to return the retired farmlands to cultivation after the subsidies ended in 2018 [42]. At the same time, reclamation cannot be avoided if the government does not assist the farmers in their non-agricultural work [43]. In addition, a new round of the GFGP was launched in 2014, although the duration of subsidies was shorter than in the first round [7]. Therefore, the result is that the area of retired farmlands from 2011 to 2020 is slightly lower than that of the previous decade.

The distribution of retired farmlands varied among regions. Besides the differences in topography and land-use patterns, the implementation of the GFGP by local governments has also led to spatial heterogeneity in land conversion. As China's poverty eradication efforts progressed, the GFGP also tilted toward serving more economically depressed areas, and thus the concentration of retired farmlands shifted from areas with better hydrothermal conditions to those areas with environmental conditions that were less suitable for farming [44]. The final vegetation type that thrives after farmland retirement depends on the local environment; thus, the policy of the GFGP was dedicated to growing grasses or forests whenever conditions would permit. Therefore, the possibility of retired farmlands that are converted into forests is greater in southern China, where water and heat conditions are good, and the possibility of retired farmlands that are converted into grasslands is greater in northern China. The effect of fallowing is evident in N due to the dense distribution of farmlands and the shift of the center of farmland reclamation [45], and the Loess Plateau region is a priority area for GFGP implementation due to severe soil erosion [46].

### 4.2. NPP of Retired Farmlands

This study showed that the distribution of NPP in retired farmlands had a pattern in the study area that was high in the southeast and low in the northwest. This pattern was consistent with the national-scale NPP patterns obtained by Sun et al. [47] and Ge et al. [48]. The high NPP in the SC and SW was attributed to the suitable environmental conditions for tree growth in the southern part of China, where most of the plantation forests were fast-growing species, such as *Eucalyptus urophylla* S.T.Blake and *Pinus armandi* Franch [49,50]. In the northern part, soil desertification, soil erosion, high evapotranspiration rates, and soil infertility caused the vegetation to have low productivity [51].

Many studies have proven that ecological restoration projects are effective in restoring and managing degraded environments [49,52]. In the present study, the dynamic changes in the NPP in retired farmlands across the country revealed an overall increasing trend and indicated positive feedback to vegetation restoration. Due to the continued cumulative increase in land vegetation biomass over time, the increase in NPP was more significant in L11–20 than in L01–20, which is the result of the GFGP implementation [53], and side by side, it shows that there is was certain negative impact at the beginning of the GFGP implementation and a certain time lag in ecological effects [54]. The extremely significant decrease in NPP was clustered in areas in N and the middle and lower reaches of the Yangtze River, where ecological restoration is ineffective. The poor results of ecological restoration efforts in these areas may be caused by the selection of fast-growing and barren-tolerant tree species that were not adapted to the local environment, or they may be the consequences of afforestation failure caused by improper early management and care [55]. Therefore, it is urgent to improve the management tools of the fragile areas of the GFGP, such as the ban on grazing in grasslands and the planting of site-specific tree species.

### 4.3. Factors for NPP Variation

Air temperature and precipitation have direct effects on the growth of vegetation [56]. Therefore, the NPP among different regions is generally positively correlated with these two factors. In arid and semi-arid regions (e.g., NW), the NPP had a weak correlation with air temperature, whereas in the wetter SC, strong correlations with air temperature were obtained. In arid and semi-arid regions, the limiting factor for the NPP is not only air temperature; that is, excessive warming is harmful to the water balance and enzyme

activity and also inhibits the photosynthetic effect [57]. Precipitation was the major driver of vegetation NPP in the arid and semi-arid regions, such as the NW [58], and it had weak correlations with the NPP in the SW. In the arid and semi-arid regions, water availability is the limiting factor because of low precipitation (<600 mm·yr$^{-1}$) and high evapotranspiration rates (annual precipitation/annual potential evapotranspiration < 0.65) [59]. Therefore, vegetation NPP in these regions was more sensitive to variations in precipitation. In the SW, high vegetation cover in the hilly areas in the SW resulted in a strong water retention capacity, so the NPP in these regions was less sensitive to the change in the precipitation rate [60]. In addition, there is a limiting effect of altitude on vegetation growth, with SW showing higher sensitivity to temperature due to high altitude [61].

In this study, the rural population density was chosen as an anthropogenic factor, because the spatial pattern of trends in population characteristics was consistent with trends in vegetation NPP [62]. The animal husbandry output was analyzed with the consideration that grazing is one of the main uses of grasslands in the northern agro-pastoral belt and is a potential cause of disturbance for the NPP [63]. Although no significant correlation was found between rural population density, livestock output value, and NPP in the regression analysis, the constraint lines showed good fits (most were quadratic curves with a single peak), which implies a good constraint relationship between anthropogenic factors and the NPP. On the left side of the curves (Figure 5), the low intensity of anthropogenic disturbance with low NPP could be caused by negative environmental conditions, such as extreme drought and soil salinization [64]. On the right side of the curve, frequent human activities such as overgrazing and agricultural activities have caused some damage to the ecosystem, and the vegetation recovery is equally poor. With a moderate level of disturbance, an increase in species' richness and biomass accumulation rate can be expected [65,66]. China's pastoral areas include Xinjiang and Tibet, where the population density is low, as well as the northern agricultural and pastoral zones, where human activities are frequent. These areas have poor soil conservation and serious grassland sanding degradation, and although some pressure was alleviated after the implementation of the GFGP, in some overgrazed areas, population growth and improper management practices have led to a tendency of secondary land degradation [67]. Therefore, in the implementation of the GFGP, decision makers should balance vegetation productivity and local farmers' livelihoods to achieve healthy long-term ecological development.

### 4.4. Limitations and Uncertainties

The retired farmlands in the present study were identified by a dataset with three phases of land use and separated into two groups: those retired within the last 10 years and those retired over 10 years ago. A finer resolution of the year of retirement than that employed in the present study could facilitate the analysis of driving forces. Moreover, the retired farmlands can be reconverted back and forth between farmland and other habitats several times, which could cause insignificance in the NPP trend and driving force analysis.

The CASA model was used to estimate the NPP from NDVI. This model has been intensively validated in China [6,68,69]. However, the CASA introduces uncertainty when implemented using a high-resolution dataset. The same vegetation type in different regions may have different characteristics and require region-specific calibration to improve model accuracy [70]. In the present study, two climate factors and two anthropogenic factors were selected for analysis. The mechanisms of factors influencing ecosystems are complex [71]. The selection of grazing intensity as an anthropogenic factor is a more realistic reflection of the relationship between grazing and vegetation growth than livestock production value. Most studies have used the number of cattle and sheep as the baseline data [72]; however, in the present study, obtaining the actual grazing intensity data proved to be difficult through the number of cattle and sheep produced because of the difference between captive- and open-stocking grazing methods in each region [73]. In future studies, more factors should be introduced for comprehensive consideration, and interactions among factors should be analyzed to explore the driving forces for vegetation NPP.

## 5. Conclusions

In this study, the retired farmlands involved in the GFGP were identified using the GlobeLand30 land-use dataset, and the NPP of the retired farmlands was estimated by the CASA model. The total retired farmland reached 261,523 km$^2$ from 2001–2020, with the largest area of retired farmlands being in the SC region of China from 2001 to 2010 and in the N region from 2011 to 2020. The NPP of retired farmlands increased significantly in most regions where the GFGP was implemented, with the highest area-specific NPP in the SW region and the lowest in N. The highest total NPP was in the SW for L01–10 and in the SC for L11–20, and the lowest was in the NW in the two groups of retired farmlands. The analysis of driving forces showed that precipitation was the dominant factor in arid and semi-arid regions, whereas temperature was more significant in high-elevation hilly regions. Significant constraint relationships existed between anthropogenic factors and the NPP, and moderate disturbances were associated with high biomass accumulation rates. This study provides details related to the spatial and temporal distribution of retired farmlands and the associated carbon sequestration capacity of those lands. The conclusions could offer critical information and guidance for the future implementation policy of the GFGP and other ecological restoration projects.

**Supplementary Materials:** The following supporting information can be downloaded at: https://www.mdpi.com/article/10.3390/land12051078/s1, Figure S1: Area of counties implementing the GFGP; Figure S2: Correlation between NPP and (a) rural population density on L01–10, (b) rural population density on L11–20, (c) animal husbandry output value on L01–10, and (d) animal husbandry output value on L11–20; Figure S3: Six regions' constraints between NPP and (a) rural population density on L01–10, (b) rural population density on L11–20, (c) animal husbandry output value on L01–10 in pastoral areas, and (d) animal husbandry output value on L11–20 in pastoral areas; Table S1: Significance test for variation trend in NPP based on β and Z; Table S2: Total NPP for vegetation on fallow land in six regions from 2011 to 2020; Table S3: Regression model statistical information.

**Author Contributions:** S.Z. and Y.Y.: Conceptualization, Methodology; Y.Y.: Supervision; Y.X. and Z.M.: Methodology, Formal analysis, Data curation, Writing—Original Draft; M.F., W.L., F.Y. and J.T.: Formal analysis, Visualization; all authors: Writing—Review and Editing. All authors have read and agreed to the published version of the manuscript.

**Funding:** This work was supported by the Young Scientists Fund of the National Natural Science Foundation of China (41901247) and 245 Qinling National Forest Ecosystem Research Station in 2022 financed by Ministry of Education of China.

**Data Availability Statement:** The data presented in this study are available on request from the corresponding author. The data are not publicly available due to the climate data being limited.

**Acknowledgments:** The authors are grateful to the National Geomatics Center of China (http://www.globallandcover.com/ accessed on 13 December 2020), the Resource and Environment Science and Data Center (https://www.resdc.cn/ accessed on 25 November 2021), the China Meteorological Data Service Center (http://data.cma.cn/ accessed on 29 October 2021), and the National Aeronautics and Space Administration (NASA) (https://www.nasa.gov/ accessed on 12 August 2022) for providing the dataset used in the study.

**Conflicts of Interest:** The authors declare no conflict of interest.

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
