# Peer review of "Analysis of Net Primary Productivity of Retired Farmlands in the Grain-for-Green Project in China from 2011 to 2020"

_land, doi:10.3390/land12051078_

Round 1
Reviewer 1 Report
Please see the attachment.

Author Response
Dear Reviewer,
We thank you for the time and effort that they have put into reviewing the previous version of the manuscript, and we appreciate the all the helpful comments and suggestions! Your suggestions have enabled us to highly improve our manuscript. According to your comments, we have made extensive modifications to our manuscript. The following are our detailed responses. We also submitted a revised version of our manuscript. Please let us know if you have other questions that need to be addressed.
Please see the attachment.
Yours sincerely,
Yan Yan

Reviewer 2 Report
The manuscript entitled “Analysis of Net Primary Productivity from Retired Farmlands in the Grain-for-Green Project in China from 2011 to 2020” is interesting and focuses on the Grain-for-Green Project (GFGP) and analysis of carbon dynamics on the retired farmlands and their driving forces. The quality of the research and the organization of the manuscript is good.
My minor concerns are as follows.
The keywords simply repeat title words and are not suitable for indexing.
Line 28: The word founding is incorrect, it should be findings.
The English language quality is insufficient and requires professional proofreading and editing. There may be spelling and grammatical mistakes creeping within the text, authors must check and correct them.
All the abbreviations should be given full form when they appear the first time in the manuscript.
The discussion largely repeats the results and needs rewriting and a “real discussion".
The English language quality is insufficient and requires professional proofreading and editing. There may be spelling and grammatical mistakes creeping within the text, authors must check and correct them.
Author Response

(The authors gave the same response as above.)

Reviewer 3 Report
The review of the manuscript titled "Analysis of Net Primary Productivity from Retired Farmlands in the Grain-for-Green Project in China from 2011 to 2020"
The research represents an evaluation of NPP in retired farmlands in China after the start of the Grain-for-Green Project (GFGP).
The introduction provides a clear description of the research problem, its background, and its current state. Methods present all necessary techniques and approaches for conducting research, including statistical methods. However, methods need a detailed explanation of "constraint relationships". How can they show the anthropogenic load? Why not use linear regression in that case, like you used it for relationships between NPP and climate factors?
Conclusions adequately describe the findings of the study. Self-citations account for less than 1% of all source literature. References need some work since quite a number of them are older than 10 years (22 (42%) of 52).
Specific comments
L27-28. Please clarify the sentence: "NPP had significant constraint relationships to rural population density 27 and animal husbandry output value". Do you mean the negative relationship or the lack of the relationship?
L36. Change the superscript to a subscript in CO2.
L46. For reader convenience, please use the same units or add the value in hectares or km2.
L101-102. I suggest abbreviating the six major regions in the same way, for example, by excluding the first letter derived from "China". For example, East China will be just "E", not "EC". The same applied to North China: ‘N’, not ‘NC’.
L175-176. Clarify the sentence "Because the effect of anthropogenic factors on NPP is not trivial, the constraint relationships between anthropogenic factors and NPP were also analyzed". What do you mean by "constraint relationships"?
L233. Fig. 3. I suggest changing the color for +1 because it looks similar to -1.
L251. Fig. 4. I suggest using the heat maps for such data.
L293. "Where there"?
L319. Provide the full biological names in italics for Eucalyptus urophylla and Pinus armandi.
L353-354. There is a contradiction in this sentence: "The effect on rural population density and livestock production value on NPP was not trivial, and no significant correlations were found in regression analysis [42].
Author Response

(The authors gave the same response as above.)

Round 2
Reviewer 3 Report
The review of the manuscript titled "Analysis of Net Primary Productivity of Retired Farmlands in the Grain-for-Green Project in China from 2011 to 2020"
This is a second review of the manuscript. The authors carefully addressed the comments of the reviewer and significantly improved the manuscript regarding clarity, typographical revision, and visual presentation. Therefore, I suggest accepting the manuscript for publication in the Land Journal after addressing the minor comments of the reviewer.
Specific comments
L105. I suggest removing the two digits after the comma; they are not necessary here because they do not make more sense. Check out the rest of the text.
L319. Provide the full biological names for Eucalyptus urophylla and Pinus armandi. For example, Eucalyptus urophylla S.T.Blake.
Author Response
Dear Reviewer,
Thank you for your important comments about our manuscript again. These comments are very helpful for improving our paper. We have studied these comments carefully and made corrections. The followings are our detailed responses. Please see the attachment. Please let us know if you have any other comments.
Yours sincerely,
Yan Yan
